# Unmet Health Needs among Young Adults with Cerebral Palsy in Ireland: A Cross-Sectional Study

**DOI:** 10.3390/jcm11164847

**Published:** 2022-08-18

**Authors:** Jennifer M. Ryan, Michael Walsh, Mary Owens, Michael Byrne, Thilo Kroll, Owen Hensey, Claire Kerr, Meriel Norris, Aisling Walsh, Grace Lavelle, Jennifer Fortune

**Affiliations:** 1Department of Public Health and Epidemiology, RCSI University of Medicine and Health Sciences, D02 DH60 Dublin, Ireland; 2Office of the Chief Clinical Officer, Health Service Executive, D20 HY57 Dublin, Ireland; 3Central Remedial Clinic, D03 R973 Dublin, Ireland; 4National Disability Children & Families Team, Social Care Division, Health Service Executive, D20 HY57 Dublin, Ireland; 5School of Nursing, Midwifery and Health Systems, University College Dublin, D04 V1W8 Dublin, Ireland; 6School of Nursing and Midwifery, Queen’s University Belfast, Belfast BT7 1NN, UK; 7College of Health, Medicine and Life Sciences, Brunel University London, London UB8 3PH, UK; 8Institute of Psychiatry, Psychology & Neuroscience, King’s College London, London SE5 8AF, UK

**Keywords:** cerebral palsy, young people, adolescents, unmet need, health services

## Abstract

Data describing the unmet health needs of young adults with cerebral palsy (CP) may support the development of appropriate health services. This study aimed to describe unmet health needs among young adults with CP in Ireland and examine if these differed between young adults who were and were not yet discharged from children’s services. In this cross-sectional study, young adults with CP aged 16–22 years completed a questionnaire assessing unmet health needs. Logistic regression was used to examine the association between discharge status and unmet health needs. Seventy-five young adults (mean age 18.4 yr; 41% female; 60% in GMFCS levels I-III) were included in the study. Forty (53%) had been discharged from children’s services. Unmet health need, as a proportion of those with needs, was highest for speech (0.64), followed by epilepsy (0.50) and equipment, mobility, control of movement and bone or joint problems (0.39 or 0.38). After adjusting for ambulatory status, unmet health needs did not differ according to discharge status. The proportion of young adults with unmet health needs highlights the importance of taking a life-course approach to CP and providing appropriate services to people with CP regardless of age.

## 1. Introduction

Cerebral palsy (CP) is a lifelong condition that primarily affects movement and function [1]. People with CP commonly experience associated conditions including sensory impairment, intellectual disability and epilepsy [1]. They may also experience complications such as pain and fatigue [2], and have a higher risk of developing chronic conditions such as cardiovascular disease and depression [3,4]. People with CP are frequent users of medical and rehabilitation services [5,6,7]. However, adults are less likely to use health services than children with CP [8,9,10].

Adults with CP report challenges accessing health services once discharged from children’s services and describe a lack of services to meet their needs [7]. A decline in service use from childhood to adulthood may therefore reflect challenges with accessing health services, rather than reduced health needs in adulthood. Indeed, the combination of a lack of services for adults with CP, challenges accessing appropriate services where they exist, and the development of complications in adulthood, may lead to increases in unmet health needs among young adults with CP. While unmet needs in areas relating to health services, education, training and social needs have previously been reported among young adults with CP [11], few studies specifically describe unmet health needs in this population.

Solanke et al. developed an unmet health needs questionnaire specifically for people with CP [12]. They defined a health need as requiring “a health service to minimise the impact of their condition or manage their functional disability” [12]. A need was considered an unmet health need if the individual reported that “the service is not provided or is not adequate” [12]. Solanke identified that adolescents with CP in the United Kingdom had significant unmet health needs in several areas; unmet needs were highest for bone and joint problems, speech and pain [12]. However, unmet health needs did not change significantly when assessed over the subsequent three years. This may be because unmet health needs were initially assessed on adolescents aged between 14 and 18 years, and the secondary effects of ageing with CP did not impact this group enough in the subsequent 3 years to cause an increase in unmet health needs.

Despite the challenges that young adults with CP experience trying to access health services once they have been discharged from children’s services [7], there is limited data describing their unmet health needs. Studies describing the unmet health needs of young adults with CP in different contexts are required to support the development of health services to meet their needs. This study aimed to describe unmet health needs among young adults with CP in Ireland and examine if these differed between young adults who were and were not discharged from children’s services. We hypothesised that young adults who were discharged from children’s services would be more likely to have unmet health needs. A secondary objective of this study was to describe health services accessed by young adults with CP in Ireland.

## 2. Materials and Methods

### 2.1. Design and Participants

We used a cross-sectional design. Young adults with CP aged 16–22 years residing in the Republic of Ireland were eligible to participate. In Ireland, people with CP typically receive healthcare from children’s services up to age 18, after which they are discharged to adult services. However, age at discharge may vary between individuals and there is no standard age, even within the same service, at which young people are discharged. We included people aged 16–22 years to capture young adults who were pre- and post-discharge. We included young adults in all Gross Motor Function Classification System (GMFCS) levels. We shared information about the study through three national organisations that provide health and social care services to people with CP; in two of these organisatons, five gatekeepers distributed 371 study invites to young people with CP living throughout Ireland. These young people received a paper version of the survey and a stamped, addressed envelope to return the survey. We also shared information about the study through disability officers in higher education, special education needs schools, professional bodies, and social media. We additionally used snowball sampling. Ethical approval was provided by Research Ethics Committees in RCSI, the Central Remedial Clinic and Enable Ireland.

### 2.2. Data Collection

We collected data using a survey. It was available online and in paper form. The survey was piloted by a person with CP and a parent. Young adults were encouraged to complete the survey alone but could complete it with support from a parent/family member/carer. Alternatively, a parent/family member/carer (hereafter referred to as parent) could complete it on behalf of the person with CP. The survey included questions relating to 1. sociodemographic and CP-related characteristics; 2. use of 11 pre-defined services; 3. use of and needs relating to assistive technology; 4. discharge from children’s services (yes/no) and if applicable, age at discharge. The survey also included a validated questionnaire that assessed unmet health needs [12]. The questionnaire asked about the following ten aspects of health care needs of people with CP: speech, mobility, positioning, equipment, pain, epilepsy, control of movement, bone or joint problems, curvature of back and eyesight. The possible responses to questions relating to speech, mobility, positioning and equipment were “help not needed”, “need more help” or “getting enough help”. Responses from questions relating to the remaining aspects were coded into the same three categories. Those reporting “getting enough help” and “need more help” were identified as having a need, and those reporting “need more help” were identified as having unmet need. As described in a previous study, the proportion of unmet needs to total needs was calculated [12].

### 2.3. Data Analysis

The distribution of data was explored using Q-Q plots for continuous data, and cross-tabulations for categorical data. Descriptive statistics (e.g., mean, frequency) were used to describe data as appropriate. Age was compared between participants who were pre-and post-discharge using an independent t-test. Remaining participant characteristics were compared between those pre- and post-discharge using Chi-squared tests. The number of services used by people who were pre-and post-discharge was compared using a linear regression adjusting for ambulatory status (i.e., GMFCS levels I–III or I–V). Assumptions of linear regression were assessed using a Q-Q plot of residuals and a scatter plot of residuals against fitted values. Logistic regression was used to compare the odds of currently accessing each service and being able to see a variety of professionals on the same day or place between those pre-and post-discharge, after adjusting for ambulatory status.

Logistic regression was used to examine associations between unmet need for each item (e.g., speech, mobility) as the dependent variable, and discharge status, GMFCS level (level I–III vs. level IV–V) and intellectual disability as independent variables. Only people with reported need for the item were included in the regression model. Unadjusted analyses were firstly conducted, with each independent variable entered separately. If more than 30 people reported need for the item, adjusted analyses were then conducted by entering independent variables into the same model.

## 3. Results

Seventy-five young adults were included. Nineteen (25.3%) questionnaires were completed by the person with CP alone, 26 (34.7%) were completed with support from a parent, and 29 (38.7%) were completed by a parent on behalf of the person with CP. Forty young adults (53.3%) had been discharged from children’s services. The mean (SD) age at discharge was 17.8 (1.2) years (range 15–20 years). Among those who were discharged, there was a mean (SD) of 2.0 (1.7) yr (range 0–6 yr) between the young person’s age at discharge and their age when they completed the survey.

Young adults are described in Table 1. The majority were male with a mean (SD) age of 18.4 (2.2) yr (range 16–22 yr). Approximately 84% had spastic CP and 60% were in GMFCS levels I–III. Type of motor abnormality differed between those who were pre- and post-discharge (*p* = 0.033); people post-discharge were more likely to have spastic diplegia or spastic quadriplegia. There was no difference in the proportion of people in GMFCS levels I–III between those pre- and post-discharge. Approximately 28% had intellectual disability, 15% had communication impairment, 15% had feeding impairment, 8% had hearing impairment and 1% had autism spectrum disorder.

All young adults lived with parents or family members. Most young adults (84%) were currently in education. Of these, 46% were in a mainstream school, 21% were in a special education needs school, 16% were in university, 8% were in a college of further education and the remainder were in other forms of education such as home schooling or vocational training. Five people were currently in employment, of which 3 were in regular part-time paid work. The five people in employment were also in education. All people pre-discharge were in education compared to 70% of people post-discharge (*p* < 0.001). Twelve people (16%) were in neither education nor employment.

Young adults resided in all regions of Ireland (Table 2). Although only 49% resided in Dublin, 80% of young adults accessed services in Dublin (Table 2).

Approximately 60% of young adults used aids or assistive technology (Table 3). Of these, the most commonly used was a manual wheelchair or electric wheelchair or scooter. Twenty per cent had an unmet need for an aid/assistive technology. There was no difference in use of or unmet need between those pre-and post-discharge.

### 3.1. Health Services

The services that people accessed are reported in Table 4. Of the 11 services listed, the mean (SD) number of services people accessed was 3.2 (2.1) (median 3; range 0–11). The mean (SD) was 3.7 (2.1) services (median 3; range 0–11) among people pre-discharge and 2.9 (2.0) services (median 2; range 0–8) among people post-discharge. People who were discharged accessed on average 1.10 (95% CI 0.20 to 2.00) fewer services than people who were not discharged after adjusting for ambulatory status (*p* = 0.017).

After adjusting for ambulatory status, people who were discharged were less likely to access occupational therapy (OR: 0.11, 95% CI 0.03 to 0.36; *p* < 0.001), speech and language therapy (OR: 0.14, 95% CI 0.04 to 0.49; *p* = 0.002), dietetics (OR: 0.10, 95% CI 0.01 to 1.00; *p* = 0.050), and assistive technology services (OR: 0.26, 95% CI 0.08 to 0.86; *p* = 0.028).

Thirty-four (45.3%) people said they could see a variety of professionals on the same day or in the same place. Pre-discharge, three people (8.6%) said this was not applicable as they were only accessing one professional compared to seven (17.5%) post-discharge. Discharge status and ability to see a variety of professionals on the same day or place were not associated after adjusting for ambulatory status (*p* = 0.132). 

### 3.2. Reported Health Needs and Unmet Health Needs

The percentage of young adults with reported health needs and unmet health needs are described in Table 5. Needs were highest for mobility, equipment and control of movement. Unmet need, as a proportion of those with needs, was highest for speech (0.64) and epilepsy (0.50). Unmet need was similar for equipment, mobility, control of movement, bone or joint problems (0.39 or 0.38). Discharge status was not associated with unmet need in unadjusted or adjusted analyses (Appendix A). In unadjusted analysis, ambulatory status was associated with unmet need relating to speech (OR: 6.00, 95% CI 1.00 to 35.9, *p* = 0.050) and eyesight (OR 12.9, 95% CI 2.22 to 74.5, *p* = 0.004), with people in GMFCS levels IV-V more likely to report unmet needs than people in levels I-III. Intellectual disability was associated with unmet need relating to mobility (OR: 3.64, 95% CI 1.10 to 12.0, *p* = 0.034). Intellectual disability and ambulatory status were not associated with unmet need in adjusted analyses.

When respondents were asked if they had any other concerns about their physical health, mental health or emotional wellbeing that their doctors, nurses and therapists had not dealt with, 28 people (39.4%) reported a concern. In response to an open-ended question asking for details about this concern(s) that were not dealt with, 16 (23%) reported a concern about mental health. Of these, 12 stated anxiety and 6 stated depression. Additionally, 6 people (8%) reported a concern relating to dental health. Other concerns reported by 1 to 3 people each were concerns with sleep, hand use, circulation, pressure sores, continence and menstrual periods. 

## 4. Discussion

This study aimed to describe unmet health needs among young adults with CP in Ireland and examine if these differed between young adults who were and were not discharged from children’s services. Unmet health need was highest for speech. More than a third of young adults reported unmet health needs in several other health areas. In contrast to our hypothesis, unmet health needs did not differ between those who were and were not discharged from children’s services.

Similar to previous reports [6,7], physiotherapy was the most commonly used service among young adults with CP. The percentage of young adults, post-discharge, who accessed occupational therapy and speech and language therapy in this study was also similar to the percentages reported in a meta-analysis of service use [7]. The relatively low percentage of young adults with access to psychology found in this study has also been reported in other studies [6,7,8]. This is the first study to our knowledge to compare if the proportion of young adults accessing services differs between those who have and have not been discharged from children’s services. Studies have compared use of services between adolescents and young adults with CP, and similarly found that non-ambulatory adolescents were less likely to use occupational therapy and speech and language therapy than non-ambulatory young adults [8,9]. However, the variation in the age at discharge found in this study highlights that it is insufficient to use age as a proxy indicator of whether a person has been discharged from children’s services if examining health and health service use of people with CP.

In a study of adolescents with CP aged 14–18 years in the UK, parents also reported the highest unmet need for speech [12]. The proportion of parents reporting unmet need for speech (0.60) was similar to the proportion we observed (0.64). Unmet needs for mobility, equipment, epilepsy, control of movement and eyesight reported in this study were higher than that reported by parents and adolescents in the UK [12]. Conversely, unmet need for pain was lower in our sample, despite the prevalence of pain being similar [12]. Unmet needs for positioning, bone or joint problems and curvature of the back reported by young adults in this study were similar to those reported by parents and adolescents in the UK [12]. However, a direct comparison of unmet health needs is challenging because of differences in characteristics of the samples and health systems. Our sample included more people in GMFCS levels IV and V. This may explain higher unmet needs given people with greater functional limitations report greater unmet need in some areas such as equipment or positioning [11,12]. However, they may report lower unmet need for pain because they are more likely to have regular access to a professional to discuss their pain with. 

We found no difference in unmet health needs between those pre- and post-discharge. Similarly, no increase in unmet health needs was observed after adolescents in the UK were discharged from children’s services [12]. This is in contrast to what we hypothesised. However, it may be because we included people up to age 22 years only or because the average time since discharge was only 2 years among those who were discharged. Adults with CP may not experience an increase in unmet health needs until they are beyond 22 years or until they are several years post-discharge, when they potentially start experiencing secondary effects of ageing with CP and have difficulties accessing appropriate services to mitigate these effects [2,7]. 

Disability is an evolving concept. As the changing, or simply ongoing, health needs of a person with CP interact with a lack of appropriate health services, they will be hindered from fully and effectively participating in society on an equal basis with others. Our findings highlight areas for service development. Young adults who were pre- and post-discharge had high unmet needs for speech and equipment. However, young adults who were discharged from children’s services were less likely to access speech and language therapy, occupational therapy and assistive technology services. These services have an important role in supporting young adults to participate in employment or training and recreational activities, and to develop social relationships and independence. Thus, provision of these services should be enhanced for young adults with CP rather than removed. The need for speech and language assessment, referral to a professional with expertise in vocational skills and independent living is outlined in clinical guidelines for adults with CP in the UK [13]. Further, people with CP living in countries that have ratified the United Nations Convention on the Rights of Person with Disabilities have a human right to services and supports that enable them to maintain full physical, mental, social and vocational ability [14]. This includes access to high-quality, affordable assistive products and professionals with the skills and knowledge to assess their need for and prescribe assistive products [14]. 

The relatively low percentage of young people accessing psychology in combination with the number of young adults who reported an unmet need relating to mental health also highlights an area for service development. Young adults with CP in Canada have similarly described challenges accessing mental health support [15]. Experiencing a mental disorder in adolescence predicts poor mental health in adulthood and can have a long-term impact on a person’s personal and economic life [16,17]. Intervening during adolescence and young adulthood, to reduce risk factors may prevent transition from subthreshold levels of anxiety/depression to comorbid mental health conditions in adulthood [18,19,20]. 

In comparison to population-based data on people with CP born in Northern Ireland between 1981 and 2001 and data from a meta-analysis of adults with CP [2,21], we believe our sample is representative of adults with CP in terms of gender, ambulatory status, and type of motor disorder. However, people with spastic unilateral CP, people with intellectual disability and people with communication impairment may be underrepresented in our sample. The cross-sectional design of this study and comparison between young adults who were and were not discharged, rather than following the same cohort over time, is a limitation of this study. However, the findings are still useful for describing unmet health needs in this population, particularly when limited data on this population currently exists in the literature. Our study is also limited by responses being collected from either the person with CP, their parent or both because their perception of unmet needs may differ; parents’ reports of unmet health needs are typically higher than adolescents’ [12]. The number of people reporting needs in some areas, particularly epilepsy, was small. Thus, the proportion with unmet needs may not be precise. Similarly, when examining associations between unmet need, GMFCS level, ID and discharge status, the small number of people reporting needs resulted in wide confidence intervals for odds ratios. Thus, any associations observed should be interpreted as areas for further exploration only. A further limitation is the lack of data describing socio-economic status of participants, which may contribute to unmet health needs.

## 5. Conclusions

In conclusion, this study confirms that young adults with CP have unmet needs in several health areas. Findings also indicate that young adults who are discharged from children’s services are less likely to access some health services. The presence of unmet health needs among young people who are pre- and post-discharge highlights the importance of taking a life-course approach to CP and providing appropriate services to people with CP regardless of age. Providing appropriate mental and physical health services to young adults with CP is not only essential for promoting health, it is essential for enabling young adults to develop independence and participate in all areas of life as they choose.

## Figures and Tables

**Table 1 jcm-11-04847-t001:** Description of young adults.

	Pre-Discharge (n = 35)		Post-Discharge (n = 40)			Total (n = 75)	
	Mean	SD	Mean	SD	*p* Value	Mean	SD
Age	16.7	0.8	19.9	1.9	<0.001	18.4	2.2
	**n**	**(%)**	**n**	**(%)**		**n**	**(%)**
Female	12	34.3	19	47.5	0.246	31	41.3
Currently in education	35	100.0	28	70.0	<0.001	63	84.0
Currently in employment	2	5.7	3	7.5	0.606	5	6.7
**Type of motor abnormality**							
Spastic hemiplegia	16	45.7	5	12.5	0.033	21	28.0
Spastic diplegia	6	17.1	15	37.5		21	28.0
Spastic quadriplegia	7	20.0	14	35.0		21	28.0
Dyskinetic	1	2.9	2	5.0		3	4.0
Ataxic	1	2.9	1	2.5		2	2.7
Don’t know or other	4	11.4	3	7.5		7	9.3
**GMFCS level ^a^**							
I	15	42.9	12	30.0	0.301	27	36.0
II	7	20.0	3	7.5		10	13.3
III	3	8.6	5	12.5		8	10.7
IV	2	5.7	5	12.5		7	9.3
V	8	22.9	14	35.0		22	29.3
**Associated impairments**							
Intellectual disability	10	28.6	11	27.5	0.918	21	28.0
Communication impairment	7	20.0	4	10.0	0.222	11	14.7
Feeding impairment	5	14.3	6	15.0	0.930	11	14.7
Hearing impairment	3	8.6	3	7.5	0.865	6	8.0
ASD	0	0	1	2.5	0.346	1	1.3
ADHD	0	0	0	0	-	0	0

^a^ n = 74.

**Table 2 jcm-11-04847-t002:** Regions of Ireland in which young people resided and accessed services.

Region (In Order of Population Density from Lowest to Highest)	Young Person (n = 75)	
	Resident, n (%)	Accessed Services ^a^, n (%)
West	3 (4.0)	3 (4.0)
Border	6 (8.0)	6 (8.0)
Midland	5 (6.7)	5 (6.7)
Mid-West	8 (10.7)	7 (9.3)
South-West	2 (2.7)	2 (2.7)
South-East	7 (9.3)	9 (12.0)
Mid-East	7 (9.3)	6 (8.0)
Dublin	37 (49.3)	60 (80.0)

^a^ Respondents may select more than one response.

**Table 3 jcm-11-04847-t003:** Use of and unmet need for aids and assistive technology.

	Discharge Status			Total, n (%)
	Pre-Discharge, n (%)	Post-Discharge, n (%)	*p* Value *	
**Using aid/assistive technology ^a^**	20 (57.1)	26 (65.0)	0.660	46 (61.3)
**Type of aid/assistive technology ^b^**				
Manual or electric wheelchair or scooter	13 (65.0)	18 (69.2)		31 (67.4)
Walking aid(s), e.g., orthopaedic footwear, walking stick, frame, rollator	7 (35.0)	14 (53.9)		21 (45.7)
Other ^c^	12 (60.0)	15 (58.0)		27 (59.2)
**Unmet need for aid/assistive technology ^d^**	7 (20.0)	8 (20.0)	0.818	15 (20.0)

^a^ n = 74. ^b^ calculated as a percentage of those using assistive technology. ^c^ includes magnifiers; large print or Braille reading materials; hearing aid(s); communication board; computer or keyboard for communicating; screen reading software; other. ^d^ n = 72. * adjusted for GMFCS level.

**Table 4 jcm-11-04847-t004:** Description of services currently accessed by young people (n = 75).

Health Service	Discharge Status			Total, n (%)
	Pre-Discharge, n (%)	Post-Discharge, n (%)	*p* Value ^a^	
Physiotherapy	26 (74.3)	28 (70.0)	0.820	54 (72.0)
Occupational therapy	28 (80.0)	15 (37.5)	<0.001	43 (57.3)
Medical	19 (54.3)	23 (57.5)	0.873	42 (56.0)
Speech and language therapy	15 (42.9)	5 (12.5)	0.002	20 (26.7)
Assistive technology	12 (34.3)	5 (12.5)	0.028	17 (22.7)
Nursing	3 (8.6)	11 (27.5)	0.145	14 (18.7)
Social work	9 (25.7)	5 (12.5)	0.106	14 (18.7)
Psychology	6 (17.1)	5 (12.5)	0.626	11 (14.7)
Personal assistance ^b^	2 (5.7)	9 (22.5)	0.142	11 (14.7)
Personal care ^c^	3 (8.6)	7 (17.5)	0.609	10 (13.3)
Dietetics	5 (14.3)	1 (2.5)	0.050	6 (8.0)

^a^ adjusted for GMFCS level. ^b^ refers to assistance provided inside and outside of the home, including personal care, support in the workplace and socialising, to enable people to live as independently as possible. ^c^ refers to support with activities of daily living (e.g., eating, dressing).

**Table 5 jcm-11-04847-t005:** Reported needs and unmet needs as a proportion of need.

Item	Pre-Discharge (n = 35)		Post-Discharge (n = 40)		Total (n = 75)	
	Reported Needs, n (%)	Unmet Need	Reported Needs, n (%)	Unmet Need	Reported Needs, n (%)	Unmet Need
Speech ^a^	15 (45.5)	0.60	10 (26.3)	0.70	25 (35.2)	0.64
Epilepsy ^b^	8 (24.2)	0.50	6 (16.2)	0.50	14 (20.0)	0.50
Equipment ^c^	22 (66.7)	0.45	29 (72.5)	0.34	51 (69.9)	0.39
Mobility	29 (82.9)	0.31	27 (67.5)	0.44	56 (74.7)	0.38
Control of movement ^d^	21 (61.8)	0.30 *	28 (70.0)	0.43	49 (66.2)	0.38 *
Bone or joint problems ^e^	16 (47.1)	0.31	23 (60.5)	0.43	39 (54.2)	0.38
Positioning ^a^	18 (54.6)	0.28	19 (50.0)	0.32	37 (52.1)	0.30
Curvature of back ^e^	12 (35.3)	0.17	18 (47.4)	0.39	30 (41.7)	0.30
Eyesight ^d^	17 (50.0)	0.20 **	23 (57.7)	0.35	40 (54.1)	0.29 **
Pain ^d^	17 (48.6)	0.18	18 (46.2)	0.12 *	35 (47.3)	0.15 *

^a^ n = 71. ^b^ n = 70. ^c^ n = 73. ^d^ n = 74. ^e^ n = 72. * denominator is number with reported needs minus one because one person with reported needs did not provide an answer for the related unmet need question. ** denominator is number with reported needs minus two because two people with reported needs did not provide an answer for the related unmet need question.

## Data Availability

The data that support the findings of this study are openly available in Zenodo at http://doi.org/10.5281/zenodo.6968034.

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
