# Peer review of "Unmet Health Needs among Young Adults with Cerebral Palsy in Ireland: A Cross-Sectional Study"

_jcm, 2022, doi:10.3390/jcm11164847_

Round 1
Reviewer 1 Report
This is an interesting and well written study building on a growing body of evidence showing a high prevalence of health needs and a smaller body of evidence showing a high prevalence of unmet health needs in youth with cerebral palsy. I have a few minor suggestions to improve the article.
1. How representative do you believe the participants are of the target population?
2. Do you know how soon post-discharge participants completed the survey? A very brief discussion addressing the system and/or recruitment efficacy would be helpful.
3. The focus on one country/health system is practical and carries some scientific advantages, but further detail about the system factors that influence the results would be helpful. Specifically, how is age at discharge from pediatric services determined? Even if there is no official policy for factors that lead to discharge, an understanding of the approach(es) would be helpful. This has a huge impact on the interpretation of all of the pre-discharge and post-discharge data. The very high percentage (70%) of the post-discharge group at higher motor impairments (GMFCS III-V) is well beyond what would be expected to be representative of the population (typically ~40% of people are GMFCS III-V). Is this group discharged at a younger age? (That would be interesting and contrary to my expectations from another country.). Is this group more likely than their post-discharged peers with lower motor impairments more likely to have received information about the survey and/or participate in the study at a higher rate?
4. The prevalence of personal care support seems to be very low. Most of the profession specific health services are easy to understand. Could you provide more detail about what is and is not “personal assistance” and “personal care”? Does this only relate to personal care through a national home care service? Does privately hired personal care support workers count? Does personal care from family or friends count?
5. I understand the rationale for the binary classification of ambulatory status. Why was this cut-off chosen? Did you look how the results are impacted if you were to split (GMFCS I & II vs GMFCS III-V)? This is more divided based on likelihood to walk in the community and may capture. It also leads to a closer balance between the two groups in your total data set.
6. Line 243-245. There is another possible explanation that could be addressed here or with more information about the recruited participants. If surveys were completed soon after discharge than an increase in unmet health needs would not be expected.
7. Line 246-247 – misworded? What is meant by “all” have high unmet needs? Only 64% of the 35% who needed it (or xx % total) have unmet needs in the results.
8. Line 284-286 – These statements are too strong for the data presented. The study “confirmed” but did not uniquely demonstrate that there are many health areas in which youth have unmet health needs. Further the first sentence implies that many youth have unmet needs in many areas, but this study does not demonstrate that. Similarly the 2nd sentence of these conclusions is too strong. There was significantly less access in less than half of the services. A comparison of access to services overall might support this conclusion, but without that data, the sentence should be reworded – potentially to “Findings also indicated that young adults who are discharged from children’s serves are less likely to access some health services”
Thank you for your interesting work.
Reviewer 2 Report
Thank you for the opportunity to review this important and interesting study.
It would be useful to have a copy of the survey included in the supplemental material. It would also be useful to know whether you felt the survey adequately identified unmet health needs or not. You mention independence and life participation in your conclusion - do you wish you had asked specifically about these?
In the results section you state "Five people were currently in employment, of which 3 were in regular part-time paid work." how were the other 2 employed?
Other studies have identified pain as a significant issue for many young people with CP but pain management has not been identified as an unmet need it your study, do you any suggestions as to why this might be?
It would be interesting to hear your thoughts on the overall impact on the young people and future health services of these health needs being unmet in the discussion. You state in your conclusion that providing appropriate services is essential for developing independence and life participation, could you expand on this in your discussion perhaps?
